# Purple sulfur bacteria fix N$_2$ via molybdenum-nitrogenase in a low molybdenum Proterozoic ocean analogue

Miriam Philippi [1], Katharina Kitzinger [1✉], Jasmine S. Berg [2], Bernhard Tschitschko [1], Abiel T. Kidane[1], Sten Littmann[1], Hannah K. Marchant [1], Nicola Storelli[3], Lenny H. E. Winkel [2,4], Carsten J. Schubert [2,5], Wiebke Mohr [1] & Marcel M. M. Kuypers[1]

Biological N$_2$ fixation was key to the expansion of life on early Earth. The N$_2$-fixing micro-organisms and the nitrogenase type used in the Proterozoic are unknown, although it has been proposed that the canonical molybdenum-nitrogenase was not used due to low molybdenum availability. We investigate N$_2$ fixation in Lake Cadagno, an analogue system to the sulfidic Proterozoic continental margins, using a combination of biogeochemical, molecular and single cell techniques. In Lake Cadagno, purple sulfur bacteria (PSB) are responsible for high N$_2$ fixation rates, to our knowledge providing the first direct evidence for PSB in situ N$_2$ fixation. Surprisingly, no alternative nitrogenases are detectable, and N$_2$ fixation is exclusively catalyzed by molybdenum-nitrogenase. Our results show that molybdenum-nitrogenase is functional at low molybdenum conditions in situ and that in contrast to previous beliefs, PSB may have driven N$_2$ fixation in the Proterozoic ocean.

[1] Department of Biogeochemistry, Max Planck Institute for Marine Microbiology, Bremen, Germany. [2] Department of Environmental Systems Science, ETH-Zurich, Zurich, Switzerland. [3] Laboratory of Applied Microbiology, Department of Environment, Constructions and Design, University of Applied Sciences of Southern Switzerland (SUPSI), Bellinzona, Switzerland. [4] Eawag, Swiss Federal Institute of Aquatic Science and Technology, Dübendorf, Switzerland. [5] Eawag, Swiss Federal Institute of Aquatic Science and Technology, Kastanienbaum, Switzerland. ✉email: kkitzing@mpi-bremen.de

As a major constituent of proteins and nucleic acids, nitrogen (N) is an essential element for life. The largest accessible N pool is in the atmosphere, where N is present in form of dinitrogen gas ($N_2$), which is chemically inert and can be used only by specialized microorganisms. Most organisms instead depend on more readily bioavailable forms of N for growth. On the early, anoxic Earth, the generation of nitric oxide and its reaction products by lightning discharge was one of the first substantial sources of readily available N[1]. Abiotic sources, however, likely did not suffice to sustain a large biosphere. Therefore, the evolution of biological $N_2$ fixation, the enzyme-catalyzed reduction of $N_2$ gas to ammonia, was key to the expansion of life on Earth[2]. $N_2$ fixation is thought to have evolved early on in Earth's history[3], with first isotopic evidence dating back to 3.2 billion years ago[4], presumably predating the evolution of oxygenic photosynthesis[5]. $N_2$ fixation is hypothesized to have evolved in an anaerobic methanogen[6] followed by horizontal transfer of the $N_2$ fixation genes to other phylogenetic groups, including cyanobacteria[6]. In modern ocean environments, $N_2$ fixation is a major source of biologically available N, counteracting local N limitation and fueling global oceanic export production[7].

The reduction of $N_2$ is catalyzed by the nitrogenase enzyme, which is composed of two major subunits: the dinitrogenase and the dinitrogenase reductase. To date, three different nitrogenase types have been described. The molybdenum iron nitrogenase (MoFe, *nif*), encoded by the structural *nifHDK* genes, is the most efficient[8] and is ubiquitous in known $N_2$-fixers[9,10]. Some $N_2$-fixers additionally encode for one or both of the alternative nitrogenases, the vanadium iron (VFe, *vnfHDGK* genes)[11] and the iron-only (FeFe, *anfHDGK* genes) nitrogenase[12]. Some studies suggest that these alternative nitrogenases evolved prior to the canonical MoFe nitrogenase and thus were responsible for N supply to the early ocean[2,13,14]. Amongst others, it has been hypothesized that MoFe nitrogenase could not have functioned due to low molybdenum (Mo) availability on early Earth[14,15]. Mo availability was likely low due to limited supply from oxygenic weathering and, later, Mo drawdown by local sulfidic conditions in parts of the Proterozoic ocean[14,15]. In contrast, recent studies using isotopic and phylogenetic analyses suggest an ancestral origin of the MoFe nitrogenase or a MoFe-related nitrogenase precursor[4,6]. In line with the latter view, experiments on $N_2$-fixing cyanobacterial cultures have shown that MoFe nitrogenase is not fully inhibited at low Mo conditions[16,17]. Yet, it is unclear if these culture-based findings for organisms that generally grow under oxic, Mo-replete conditions can be directly transferred to environmental organisms that generally live under anoxic, Mo-limiting conditions.

These diverging theories have sustained considerable debate about the origin of nitrogenases, the organisms involved, as well as when and how biologically fixed N became broadly available in the ancient ocean. These questions are difficult to address, as Precambrian samples are sparse and their informational content is restricted to bulk N-isotope composition[4,13] or, in rare cases, biomarkers attributed to potential $N_2$-fixing microorganisms[18]. Isotope composition provides indications for $N_2$ fixation activity and which nitrogenase type might have been active, without revealing the identity of the responsible organisms. In contrast, biomarkers provide evidence toward identity but cannot be used to deduce the activity of potential $N_2$-fixers. Therefore, it is almost impossible to link the identity and function of microorganisms using only paleontological data for these ancient samples. However, present-day analogue systems can provide insights into how these processes might have occurred in the ancient ocean and which organisms might have been responsible.

For the Proterozoic ocean, it is often assumed that cyanobacteria performing oxygenic photosynthesis were the key primary producers and $N_2$-fixers[19], although direct evidence linking cyanobacteria to ancient ocean N-input is lacking. In addition to cyanobacteria, it has been speculated that anoxygenic phototrophs may have locally contributed to both $CO_2$ and $N_2$ fixation[17,20]. In the late Proterozoic ocean, euxinic (i.e., anoxic and sulfidic) conditions persisted at the continental margins and intruded into the photic zone, below a mildly oxygenated surface[18,21–23]. The resulting co-availability of light and sulfide provided ideal conditions for anoxygenic phototrophic sulfur bacteria. Members of both the green sulfur bacteria (GSB) and the purple sulfur bacteria (PSB) have the genetic potential for $N_2$ fixation e.g.[9] and their $N_2$ fixation activity has been shown in pure cultures[24]. Fossil evidence for anoxygenic phototrophic sulfur bacteria (GSB and PSB) indicates that these organisms date back at least 1.6 billion years[18] and therefore, they could have contributed to $N_2$ fixation in the Proterozoic.

While $N_2$ fixation activity has been demonstrated in the environment for GSB at low rates[25,26], in situ observations of $N_2$ fixation by PSB are missing. The genetic potential for $N_2$ fixation is present in several environmentally relevant PSB e.g.[27–29] but genetic potential alone cannot be unambiguously translated to activity in the environment[30].

Lake Cadagno is commonly considered an analogue system for the continental ocean margins during the Proterozoic (2.5–0.5 billion years ago) due to similar environmental conditions e.g.[31–34]. These include a permanently stratified water column with euxinic bottom waters, oxygenated waters restricted to the surface, a shallow chemocline located in the photic zone, low sulfate concentrations[31] and Mo concentrations ($<10$ nmol $L^{-1}$)[33] that are at least an order of magnitude lower than modern ocean Mo concentrations ($\sim105$ nmol $L^{-1}$)[35]. The low Mo conditions of the chemocline are expected to favor the use of alternative nitrogenases[10], while $N_2$ fixation by cyanobacteria using the conventional MoFe nitrogenase would be expected in the oxygenated surface waters, where Mo concentrations are slightly higher[33].

Here, we investigate the organisms responsible for $N_2$ fixation and the nitrogenase types used in the anoxic, low Mo chemocline of Lake Cadagno, by combining natural abundance isotope measurements with stable isotope incubations, metagenomics, metatranscriptomics, and single-cell analyses. We show that despite low Mo concentrations, diverse PSB using conventional MoFe nitrogenase are the key $N_2$-fixing microorganisms in Lake Cadagno. Our results imply that PSB may have contributed to N availability in the Proterozoic ocean.

## Results and discussion

**Chemocline physicochemical parameters and biomass $\delta^{15}N$.** In August 2018, the chemocline (defined as the zone of constant conductivity) of Lake Cadagno was situated between 13.5 m and 14.5 m water depth (Fig. 1), which is consistent with previous observations for this season in recent years[36]. Mo concentrations throughout the investigated water column were consistently low, $<10$ nmol $L^{-1}$ (Table S1, Fig. 1a), as previously reported for Lake Cadagno[33]. These Mo concentrations are in the range proposed for the Proterozoic ocean[37,38]. Sulfide and nutrient concentrations were similar to earlier years[25,26], with high concentrations of up to 80 μM sulfide and 30 μM ammonium in the bottom waters (Fig. 1a, b, Table S1). Sulfide concentrations were at or below the detection limit in the chemocline and increased sharply just below. In contrast, ammonium diffused upwards into the chemocline where it eventually disappeared due to microbial consumption (Fig. 1b). A pronounced turbidity maximum coincided with the chemocline (Fig. 1d), indicating high cell densities.

We observed low $\delta^{15}N$ values of $\sim1$ to 2.5‰ for bulk biomass in the chemocline (Fig. 1b, Table S2). Such low values are

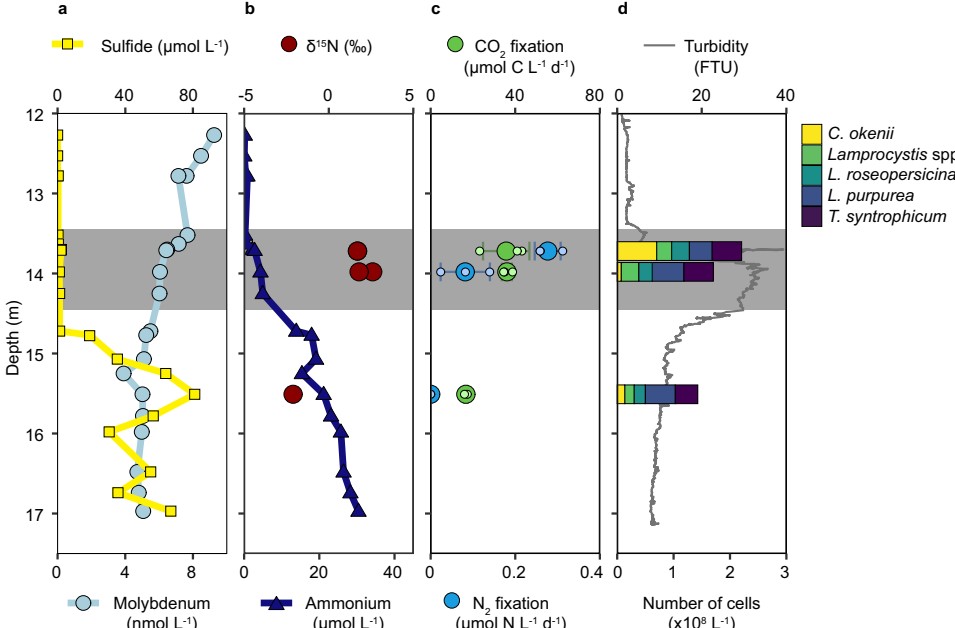

**Fig. 1 Lake Cadagno chemocline nutrient, δ[15]N and turbidity depth profile, bulk CO₂ and N₂ fixation rates, and PSB abundance. a** Molybdenum (light blue) and sulfide (yellow) depth profiles. **b** Ammonium (dark blue) and δ[15]N values of bulk biomass (red circles, duplicate measurements at depths 13.7 m and 14 m, single measurement at 15.5 m). **c** Bulk N₂ fixation rates (green) and bulk CO₂ fixation rates (blue, determined in the same samples as N₂ fixation rates). All rate measurements were performed in biological triplicates (except 15.5 m where one replicate was lost), individual replicates are shown as small circles, and the average rate is shown in large circles with the respective standard deviation. N₂ fixation rates at 15.5 m depth were below the detection limit. **d** Abundances of key PSB populations (stacked bars, cell numbers based on FISH counts) and turbidity (gray line, a measure of total cell abundance). The location of the chemocline, as determined from the conductivity profile, is indicated by gray shading (**a–d**).

indicative of N input through N₂ fixation[39], although external reduced N with low δ[15]N signatures from the oligo- to mesotrophic underwater springs and surface-runoffs[40] might have contributed. Below the chemocline, δ[15]N values were even lower (∼−2‰, Fig. 1b, Table S2). These lower values could also stem from N₂ fixation, or alternatively, from preferential uptake of isotopically light ammonium under ammonium replete conditions[41]. These isotopic signatures, taken together with reports of N₂ fixation activity from earlier years[25,26], indicate that N₂ fixation constitutes a persistent feature of the Lake Cadagno chemocline.

**Bulk N₂ and CO₂ fixation rates in the chemocline.** Bulk community CO₂ and N₂ fixation rates were determined at two depths within, and one depth below the chemocline (Fig. 1c). N₂ fixation rates were highest in the upper part of the chemocline (0.27 μmol N L⁻¹ d⁻¹), about half as high mid-chemocline, and were undetectable below. The chemocline rates were much higher than those previously reported for Lake Cadagno[25,26] and the meromictic Lake Malawi[42]. They were comparable to the highest N₂ fixation rates reported for the marine environment[43] and the brackish Baltic Sea e.g.[44,45], all of which are attributed to cyanobacteria.

Lake Cadagno N₂ fixation rates were negatively correlated with ammonium concentrations. Fixation rates were highest in the shallowest samples, where ammonium concentrations were lowest, and decreased with depth to below detection limit below the chemocline, where ammonium concentrations were highest. The detection of N₂ fixation activity in the presence of ammonium is in line with previous findings from Lake Cadagno[25,26]. Yet, single-cell analyses indicate that *Chlorobium phaeobacteroides*, the only confirmed in situ N₂-fixer in Lake Cadagno, appears not to fix N₂ in the presence of ammonium[26],

suggesting that N₂-fixers other than this GSB might have been active in our incubations.

CO₂ fixation rates in the two upper depths located within the chemocline were similar, while the rate underneath was lower. These rates were comparable to previous rates of CO₂ fixation reported for the chemocline of Lake Cadagno[29,36,46]. Assuming that all assimilated inorganic C has to be matched by N assimilation in the C/N ratio of the bulk biomass, we calculated depth-specific autotrophic N demands, defined as the amount of N required for the respective autotrophic C-based growth. The autotrophic N demand exceeded the measured N₂ fixation rates at all depths, which is commonly the case in the environment, as the predominant N source for autotrophic growth is regenerated (recycled) N like ammonium and/or nitrate[47]. Further, N₂ fixation rates did not make up a constant fraction of the autotrophic N demand. The contribution of N₂ fixation varied with depth, with N₂ fixation supporting 7.3%, 1.9% and 0% of the measured autotrophic N demand at 13.7 m, 14 m, and 15.5 m depth, respectively.

Within the chemocline, the calculated average autotrophic N demand was ∼4.2 μmol N L⁻¹ d⁻¹. Assuming that the chemocline is constantly mixed[48] and that ammonium diffusing upwards into the chemocline is therefore also mixed across this layer, we calculated an ammonium flux of 3.5 μmol cm⁻² d⁻¹ into the chemocline. This translates into an ammonium assimilation rate of 3.5 μmol N L⁻¹ d⁻¹ in the chemocline (13.45 m to 14.45 m depth), which matches previous experimentally determined in situ ammonium assimilation rates[46] and flux-based estimates[25]. Therefore, in addition to N₂ fixation, ammonium assimilation was an important N source for growth in the chemocline, sustaining ∼80% of the autotrophic carbon fixation.

**Meta-omics reveal PSB as candidates for N₂ fixation.** To assess which microorganisms could be responsible for the high

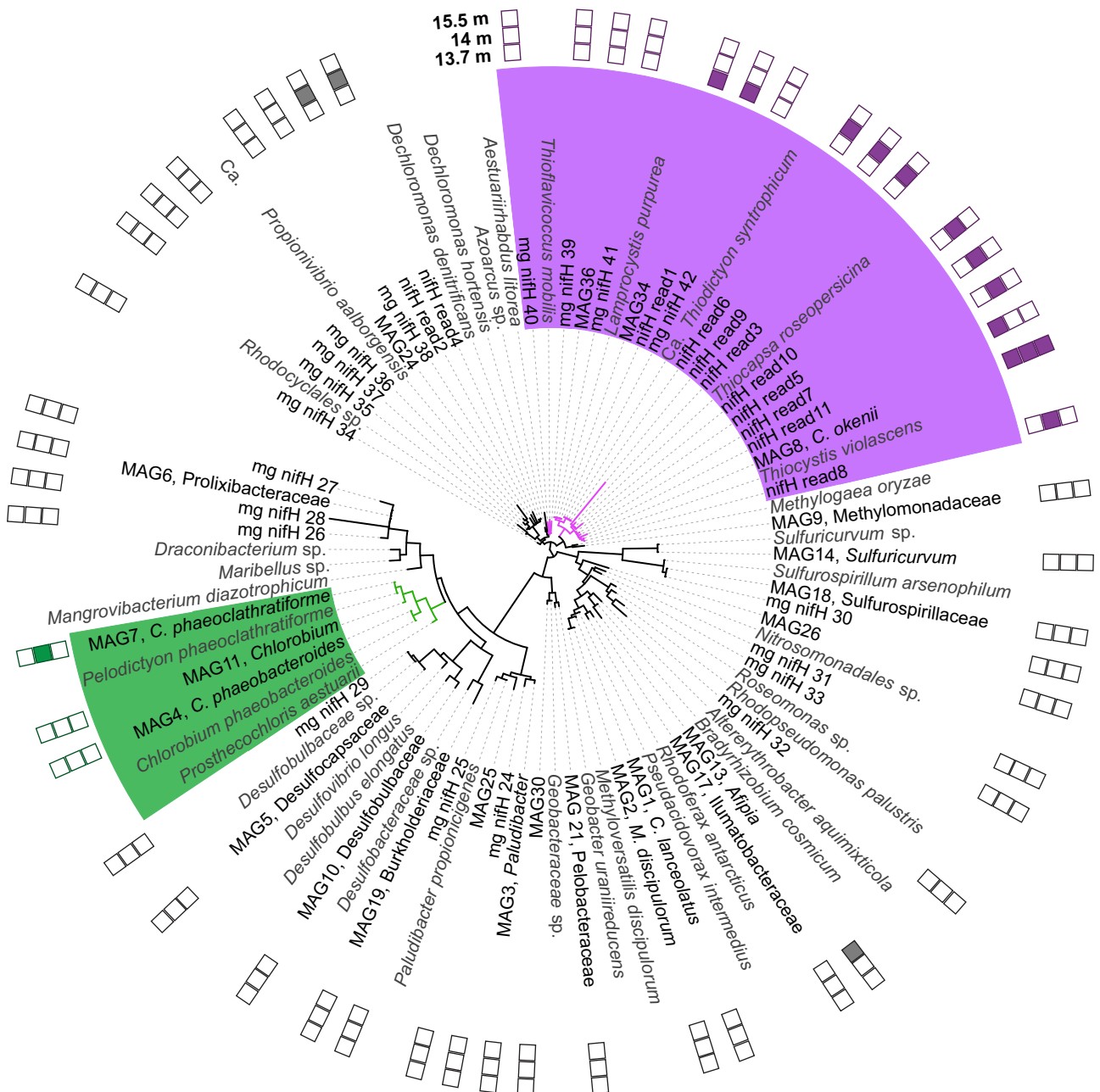

**Fig. 2 NifH diversity in Lake Cadagno metagenomes and metatranscriptomes.** Tree shows all verified NifH amino acid sequences obtained from the assembled metagenome from 2014 (MAG sequences and unbinned mg_nifH sequences) and the three metatranscriptomes from 2018 (nifH_read sequences). Reference sequences obtained from NCBI are indicated in gray. Transcription (presence) is indicated by filled boxes in 13.7 m, 14 m, and 15.5 m water depths from inner to outer circle, respectively, while empty boxes indicate that no transcription of the gene was detected. Detailed read counts are listed in Table S3. Sequences originating from GSB and PSB are highlighted in green and purple, respectively. MAG taxonomy was inferred from GTDB-Tk classification. The lowest taxonomic rank assigned is shown, with no taxonomic information indicating that classification based on GTDB-Tk was not possible due to a lack of sufficient marker genes (see also Fig. S1).

measured N₂ fixation rates and specifically, which role anoxygenic phototrophs play, we analyzed metagenomic data from Lake Cadagno sampled in 2014[29]. In our analyses, we focused on metagenome-assembled genomes (MAGs) encoding for the structural genes of the nitrogenase enzyme and unbinned *nif* (as well as *vnf* and *anf*) genes.

Due to the sulfidic, low Mo conditions in the Lake Cadagno chemocline, one might expect the use of alternative nitrogenases rather than MoFe nitrogenase[10,14]. However, we could not detect

genes encoding for any of the subunits of the alternative VFe or FeFe nitrogenase (*vnf*, *anf*) in the metagenome data. Instead, we observed a high diversity of potential N₂-fixers encoding for MoFe nitrogenase, with 36 *nif* gene-encoding MAGs from at least nine different phylogenetic classes, including PSB, GSB, sulfur-reducing bacteria, and several Burkholderiales (Fig. S1). The high diversity of potential N₂-fixers was visible from NifH as well as NifDK sequences, which together encode the structural components of the nitrogenase enzyme (Fig. 2, S3, S4). While PSB and

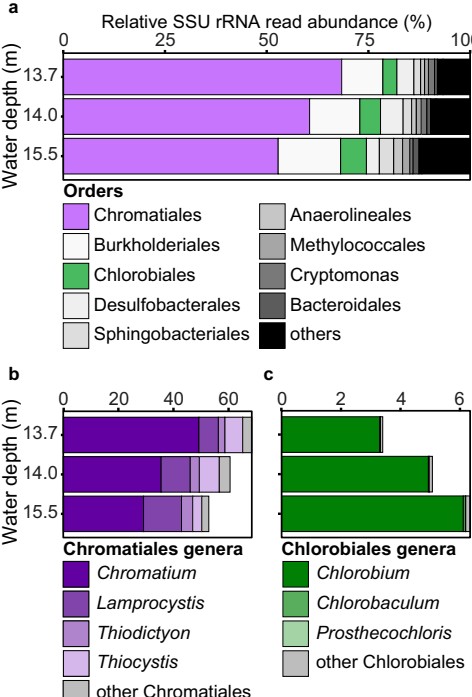

**Fig. 3 In situ abundance and activity profile of the microbial community in Lake Cadagno, based on relative small subunit rRNA read abundance in 2018. a** Relative small subunit rRNA read abundance of the entire microbial community in Lake Cadagno chemocline classified to order level. Chromatiales (PSB) are indicated in purple, Chlorobiales (GSB) in green. **b** Relative small subunit rRNA read abundance of Chromatiales genera. **c** Relative small subunit rRNA read abundance of Chlorobiales genera.

GSB NifH/D/K were only distantly related to each other, sequences belonging to members of these groups clustered together (with the exception of the NifH sequence of the PSB *Thioflavicoccus*). Interestingly, all PSB genomes sequenced so far exclusively encode MoFe nitrogenase[9,49], even though the euxinic and thus low Mo conditions in which they thrive should favor alternative nitrogenases[10,14].

In the Lake Cadagno metagenome, two *nif*-containing MAGs were classified as PSB (Chromatiaceae), namely *Chromatium okenii* and *Lamprocystis purpurea* (Fig. S1). *C. okenii* comprised a large part of the chemocline biomass (Fig. S2)[48] and its genetic potential for $N_2$ fixation has previously been described[28,29] (for discussion of the *nif* genes found in *C. okenii*, see Supplementary Note S1). In addition, we identified a partial, unbinned *nifH* gene in the metagenome (mg_nifH_42) that closely resembled the *nifH* gene of *Thiodictyon syntrophicum* (Fig. 2), isolated from Lake Cadagno and encoding a full set of *nif* genes[27].

We further identified four *nif*-containing GSB MAGs (Chlorobiaceae), all belonging to the genus *Chlorobium*. One of these was classified as *C. phaeobacteroides*, which is the only $N_2$-fixer in Lake Cadagno with confirmed in situ $N_2$ fixation activity[25,26]. Another MAG was classified as *Chlorobium clathratiforme*, for which in situ expression of $N_2$ fixation genes has been shown in Lake Cadagno[50].

Likely as an adaptation to the low Mo conditions, all PSB and GSB MAGs containing MoFe nitrogenase encoded for high affinity molybdate ABC transporter genes (with the exception of the GSB *C. phaeobacteroides*).

To assess which of the potential $N_2$-fixers identified in the metagenome from 2014 could be involved in active $N_2$ fixation in the chemocline of Lake Cadagno, metatranscriptomes from our

study in 2018 were sequenced. Taxonomic affiliation of the transcribed *nifH* genes revealed that the majority of all *nifH* transcripts originated from PSB (Table S3, Fig. 2). Most *nifH* reads were assigned to *C. okenii*, which was the only population found to transcribe *nifH* at all three investigated depths (Fig. 2, Table S3). Transcription of PSB *nif* genes was consistent with measured bulk $N_2$ fixation rates, with most *nif* reads obtained from the upper two depths (Table S3), where significant $N_2$ fixation was detected in situ (Fig. 1). Fewer *nif* transcripts were obtained from below the chemocline, where no $N_2$ fixation rate was measurable, despite this sample having the largest sequencing depth (Tables S3 and S4). Corroborating the findings from the metagenomes, no transcripts for alternative nitrogenases were found at any depth. This further indicates the dominance of MoFe nitrogenase-based $N_2$ fixation in this system. Interestingly, the high-affinity molybdate ABC transporters encoded by the obtained PSB MAGs were transcribed at the time of our study (Supplementary File S1), likely enabling PSB to acquire Mo even at the low Mo concentrations in the Lake Cadagno chemocline.

Only a few GSB-related *nifH* transcripts were found in the metatranscriptomic dataset (Table S3, Fig. 2), which is in contrast to previous *nifH* transcription analysis[25]. These findings indicate that PSB, especially *C. okenii*, may have been more important for $N_2$ fixation in the chemocline of Lake Cadagno at the time of sampling than any other potential $N_2$-fixer, including GSB.

As a combined proxy for abundance and general cellular activity, we further investigated the distribution of small subunit (SSU) rRNA reads throughout the transcriptome samples (Fig. 3). In addition to low transcription of *nif* by GSB in 2018, comparably little contribution to the total SSU reads by GSB was found (Fig. 3a, c). Relative GSB SSU read abundance increased with depth, likely due to the adaptation of GSB to low-light conditions[51].

We found a clear dominance of PSB based on SSU reads in all three investigated depths in 2018 (Fig. 3). PSB contributed >50% of all SSU reads in all depths, with the highest relative abundance (>60%) at the shallowest depth, where $N_2$ fixation rates were highest. The largest fraction by far of SSU reads was assigned to the genus *Chromatium* (>25% in all samples), followed by the genus *Lamprocystis* (Fig. 3b). Two additional abundant and active PSB genera, *Thiodictyon* and *Thiocystis*, were detected based on relative SSU read abundance (Fig. 3b).

**PSB single-cell activity and contribution to a bulk rate.** Based on the metagenomic and metatranscriptomic findings, five populations of PSB with the genetic capacity to fix $N_2$ were visualized and quantified using fluorescence in situ hybridization (FISH): *Chromatium okenii*, *Lamprocystis* spp., *Lamprocystis roseopersicina*, *Lamprocystis purpurea*, and *Thiodictyon syntrophicum* (Fig. 1d). All five investigated PSB populations were detected at all three incubation depths, and at similar abundances, as previously reported for Lake Cadagno[46]. Together, the investigated PSB accounted for around 15–20% of the microbial community (Fig. 1d, total cell numbers were $1.01 \times 10^9 \pm 0.03 \times 10^9$; $1.15 \times 10^9 \pm 0.16 \times 10^9$, and $7.90 \times 10^8 \pm 0.83 \times 10^8$ L$^{-1}$ for 13.7 m, 14 m, and 15.5 m water depth, respectively, see Supplementary File S1). Cell numbers of the *Lamprocystis* populations and the *T. syntrophicum* population were comparable in the upper two depths located in the chemocline and decreased below. In contrast, *C. okenii* cell abundance in the chemocline was variable, with the highest abundance in the shallowest sample, possibly due to phototactic movement[28]. Overall, total cell numbers of the investigated PSB declined with depth (Fig. 1d), following the same trend as the relative abundance of PSB SSU

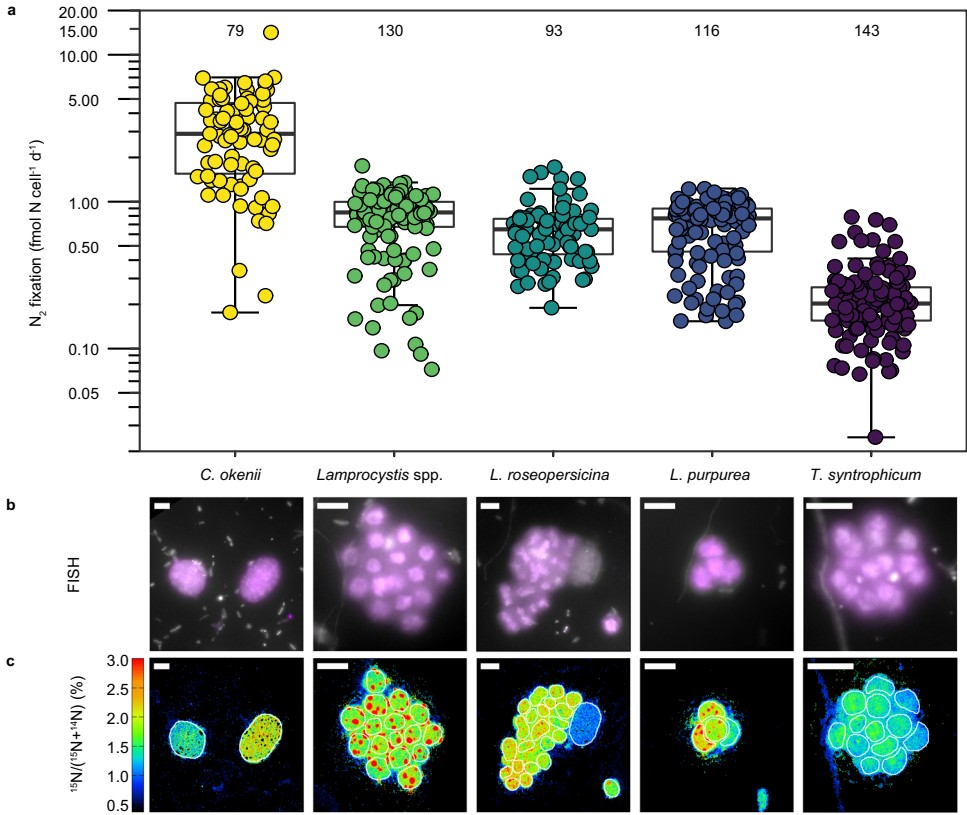

**Fig. 4 Single cell N$_2$ fixation activity determined by nanoSIMS. a** Single-cell N$_2$ fixation rates of individual PSB populations in 13.7 m water depth. The number of cells analyzed per population (*n*) is shown above each box plot. Boxplots depict the 25–75% quantile range, with the centerline depicting the median (50% quantile); whiskers encompass data points within 1.5× of the interquartile range. *Y*-axis is in log scale. **b** Representative epifluorescence and **c** nanoSIMS images of the individual PSB populations. Scale bars (**b** and **c**) represent 4 µm. Individual target PSB populations (in purple) were identified using specific FISH probes. All cells were counterstained with DAPI (white). PSB in nanoSIMS images are outlined in white. Note the presence of one *C. okenii* cell in the *L. roseopersicina* images. For each population, four independent FISH experiments were conducted, with similar results as shown in panel (**b**).

reads in the metatranscriptomes (Fig. 3a, b), the N$_2$ fixation activity, and bulk cell density (Fig. 1c, d).

To measure in situ N$_2$ fixation activity by the five PSB populations and to evaluate their specific contribution to bulk activity, single-cell analysis using nanoscale secondary ion mass spectrometry (nanoSIMS) was performed.

The uptake of $^{15}$N (from $^{15}$N$_2$) and $^{13}$C (from $^{13}$CO$_2$) into individual PSB and surrounding cells (Figs. S5 and S6) was measured in 13.7 m water depth, using the replicate where the bulk N$_2$ fixation rate was highest. All investigated PSB populations were significantly enriched in both $^{15}$N (Fig. 4, S5) and $^{13}$C (Fig. S6), with the exception of one *Lamprocystis* spp. cell that was neither significantly enriched in $^{15}$N nor in $^{13}$C, indicating that this cell was physiologically inactive or dead. To our knowledge, this provides the first direct evidence for N$_2$ fixation by PSB in situ. Overall, the three *Lamprocystis* populations had the highest cellular $^{15}$N enrichments, while enrichment of *T. syntrophicum* and *C. okenii* cells was slightly lower (Fig. S5). Interestingly, the analyzed *Lamprocystis* cells contained distinct subcellular areas of high $^{15}$N enrichment (Fig. 4c), similar to previous observations of N$_2$-fixing cyanobacterial cells[52], indicating storage of freshly fixed N. Taking into account the biomass and biovolumes of the PSB (Fig. S7), population-specific single cell N$_2$ fixation rates ranging from 0.23 to 3.29 fmol N cell$^{-1}$ d$^{-1}$ were calculated. These rates are at the lower end of single-cell rates reported for marine cyanobacteria e.g.[53]. Yet, they were notably higher than the single-cell rates

reported for *C. phaeobacteroides* in Lake Cadagno (0.0–0.02 fmol N cell$^{-1}$ d$^{-1}$)[25] and a marine non-cyanobacterial N$_2$-fixing culture (up to 0.09 fmol N cell$^{-1}$ d$^{-1}$)[30].

Although PSB were actively fixing N$_2$, the freshly fixed N did not meet the total PSB N demand for growth (on average 27.7% to 83.3%, determined from the measured single-cell CO$_2$ fixation and bulk C/N ratios, see Fig. S8). This is consistent with the measured bulk CO$_2$ fixation rates, which significantly outpaced N$_2$ fixation, and indicates the uptake of an alternative N source, even within the diazotrophic community. In situ ammonium assimilation of *Chromatium* and *Lamprocystis* has been demonstrated on a single cell level[46] and simultaneous N$_2$ fixation activity and ammonium uptake has previously been shown for a *Klebsiella* culture[54]. The additional N demand of the PSB was therefore likely met by assimilation of upward diffusing ammonium. PSB N$_2$ fixation might further increase with decreasing ammonium concentrations, as previously observed for *Klebsiella*[54].

All of the investigated PSB are large in comparison to the majority of other microbial cells in Lake Cadagno, with *C. okenii* being by far the largest of the five investigated PSB populations (Fig. S7). This is reflected in very high *C. okenii* per cell N$_2$ fixation rates, which were five to fourteen times higher than the per cell rates of the other investigated PSB, even though *C. okenii* cells were less enriched in $^{15}$N than *Lamprocystis* cells (Fig. S5). Taking into account the in situ cell abundance and the average per cell N$_2$ fixation rates, we calculated the contribution of the

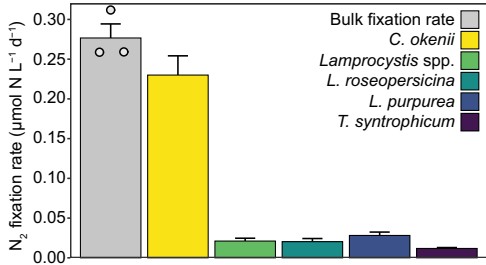

**Fig. 5 Total N$_2$ fixation rates of investigated PSB in comparison to bulk N$_2$ fixation rate in 13.7 m water depth.** The bulk rate is shown as the average of the triplicate incubations, with the error bar representing the standard deviation. Gray circles represent the individual replicates. Population-specific rates were calculated based on the average cell abundance and average single-cell N$_2$ fixation rate. Here, error bars represent the propagated standard error of the cell abundance and the single-cell N$_2$ fixation rate.

different PSB populations to the measured bulk N$_2$ fixation rate (Fig. 5). We found that *C. okenii* alone was responsible for >80% of the average bulk N$_2$ fixation rate. Also in a replicate experiment conducted the following day, *C. okenii* contributed >80% of the bulk N$_2$ fixation rate, showing that this microorganism consistently drives N$_2$ fixation in the Lake Cadagno chemocline (Supplementary Table S5, Supplementary File S1). This is in contrast to an earlier study that attributed N$_2$ fixation in the Lake Cadagno chemocline mainly to the GSB *C. phaeobacteroides*[25], however, based on a limited dataset. PSB were not investigated in this earlier study and may have contributed to bulk N$_2$ fixation. In our dataset, taken together, the investigated PSB populations could account for the total bulk N$_2$ fixation activity (99.8 ± 11.9% of the sample analyzed with nanoSIMS; see Supplementary Note S2), making PSB the key N$_2$-fixers in the Lake Cadagno chemocline at the time of our study.

**Implications for N$_2$ fixation in the Proterozoic ocean.** Lake Cadagno is frequently used as an analogue system for the euxinic continental ocean margins during the Proterozoic eon (2.5–0.5 billion years ago) due to the assumed similar environmental conditions e.g.[31–34]. These include a stratified water column with euxinic bottom waters, oxygenated waters restricted to the very surface layers, a shallow and photic chemocline, low sulfate concentrations[31], and low Mo concentrations[33]. Mo concentrations in the Lake Cadagno chemocline are consistently below 10 nmol L$^{-1}$ (Fig. 1a, Table S1)[33], and thus within the proposed range for the Proterozoic ocean[37,38]. It has been proposed that Mo limitation in the Proterozoic ocean hampered N$_2$ fixation by the conventional MoFe nitrogenases and that alternative nitrogenases were active instead[14]. Alternatively, it has been hypothesized that even though N$_2$ fixation activity by cyanobacterial MoFe nitrogenase decreases with decreasing Mo concentrations[17], the low Mo conditions of the Proterozoic ocean might not have fully inhibited MoFe nitrogenase activity[3,17]. It is, however, unclear if the conclusions from these cyanobacteria culture-based findings[17] can be generally transferred to N$_2$-fixers in complex environments with constant Mo limitation.

Although Mo concentrations are low in Lake Cadagno, we did not detect alternative nitrogenases. Instead, we show that PSB encoding only MoFe nitrogenases actively fix N$_2$ under low Mo conditions and contribute substantially to the N inventory of Lake Cadagno. This is in line with the cyanobacteria culture-based findings[16,17] and challenges the theory that activity by MoFe nitrogenases is inhibited under low Mo conditions as

proposed for the Proterozoic ocean. It further conforms to more recent studies suggesting an ancestral origin of the MoFe nitrogenase or a MoFe-related nitrogenase precursor in comparison to the VFe or FeFe nitrogenases[4,6].

It is often assumed that the majority of new N input into the Proterozoic ocean stemmed from N$_2$-fixing oxygenic photo-trophs, the cyanobacteria[17,19]. While anoxygenic phototrophs, like GSB and PSB, have been implicated as important primary producers in euxinic and nutrient-rich continental margins of the Proterozoic ocean[18,20,55], they have so far been underappreciated as potential N$_2$-fixers[17]. We found that, under environmental conditions considered similar to those in the ancient ocean, PSB are the main N$_2$-fixers and thereby add a substantial amount of new N to the ecosystem. GSB have also previously been found to fix N$_2$ in this Proterozoic analogue, albeit, at significantly lower rates[25,26]. PSB and GSB have different light adaptations[51]: GSB are often adapted to lower light levels and thus grow at greater depth. Moreover, the recently discovered ability of PSB to couple C fixation to oxygen consumption[29] could suggest that PSB were more abundant in shallower waters of the Proterozoic ocean, at the transition zone of sulfidic to mildly oxygenated surface waters. In combination with the higher growth rates and activity of PSB in comparison to GSB[46], we hypothesize that N$_2$ fixation by PSB was a substantial source of fixed N, particularly for the photic surface waters of the Proterozoic ocean.

While PSB single-cell N$_2$ fixation rates in Lake Cadagno were high, N$_2$ fixation activity alone could not account for the average PSB autotrophic N demand (Figure S8) but supplemented the N demand that could not be met by ammonium[46]. As such, the measured PSB N$_2$ fixation rates might not reflect their maximum N$_2$ fixation capacity, and, as is the case for other N$_2$-fixers[54], PSB N$_2$ fixation might be even higher under more N-limited conditions, such as those found on early Earth[3]. Our combined data suggest that PSB could have tolerated more N- and Mo-limited conditions in the Proterozoic ocean than previously thought[20,55]. Furthermore, our results imply that N$_2$ fixation by PSB could have been a significant source of new, bioavailable N in euxinic photic waters.

## Methods

**Sampling.** Samples were collected on 28 August 2018 during a field campaign to Lake Cadagno[29], Switzerland. In situ measurements and water collection was performed at the deepest part of the lake (21 m). Water was collected using a pump CTD system as described in Di Nezio et al.[36]. Online in situ data were obtained during a continuous downcast of the CTD-system from the water surface down to ~17.5 m depth. During the upcast, discrete water samples were collected from a total of 20 depths (between 12 m and 17 m) above, in, and below the chemocline for chemical analyses and from 3 depths for incubation experiments (13.7 m, 14 m, and 15.5 m).

In Lake Cadagno, wind-driven internal waves lead to vertical shifts of the water masses and their corresponding physicochemical parameters[56]. While sampling, it was apparent that the depths of the individual water masses had slightly shifted between the down- and the upcast. Therefore, we corrected the water depths of the samples collected during the upcast so that the physicochemical parameters during sampling best matched those of the continuous downcast, to ensure that samples were assigned to the respective water mass that they originated from. A custom R script was employed for the depth correction. In brief, all parameters measured by the CTD-system during the upcast and the downcast were normalized to percent (with 100% as the maximum observed value, and 0% the minimum observed value). Per individual sampling depth (during the upcast, where the pump cast CTD remained stationary for some time), average values of conductivity, temperature, and pressure were calculated and converted to percent values. Then, the depth from the downcast profile was identified that best matched all calculated percent values. This was achieved by subtracting the percent values per parameter from all respective data points of the downcast profile. Absolute values of the calculated differences per data row were summed. The depth with the lowest resulting sum, i.e., with the most similar physicochemical parameters, was then chosen as the corrected depth.

**Chemical analyses, flux calculations, and rate determinations.** For chemical analyses, lake water from the individual sampling depths was sterile-filtered (0.2 μm, cellulose acetate filter) and frozen at −20 °C until analysis. Samples were analyzed with a QuAAtro39 autoanalyzer (Seal Analytical) using the methods

described in Strickland and Parsons[57] to determine concentrations of dissolved inorganic phosphorus ($PO_4^{3-}$), nitrite ($NO_2^-$), nitrate ($NO_3^-$), and reactive silica (Si). Ammonium concentrations were determined from the same filtered samples using the colorimetric analysis described in Kempers et al.[58]. Molybdenum concentrations were determined from filtered samples after acidification with 1% $HNO_3$ (69%, ROTIPURAN®, Roth) using an ICP-MS 7900 (Agilent, Santa Clara, USA). Molybdenum was analyzed on mass 95 in He-mode using a multi-element calibration SRM (21 elements, Bernd Kraft). The SRM NIST 1643f was analyzed in parallel to guarantee the quality of analyses. Concentrations of sulfide were determined colorimetrically from unfiltered Lake water samples, following Cline[59].

To calculate the turbulent flux ($J$) of ammonium into the chemocline, we assumed a steady-state using Fick's first law: $J = -D\partial C/\partial x$. A turbulent diffusion coefficient ($D$) of $1.6 \times 10^{-6}$ m$^2$ s$^{-1}$ was used, corresponding to turbulence at the Lake Cadagno chemocline boundaries[60]. The change in concentration ($\partial C$) was calculated over 14.25 m to 14.77 m depth, where the steepest ammonium gradient was observed. Ammonium uptake rates were calculated for the chemocline by integrating this flux over the chemocline from 13.45 m to 14.45 m depth.

To quantify $N_2$ fixation and primary production (i.e., $CO_2$ fixation) rates, stable isotope incubations with $^{15}N_2$ and $^{13}CO_2$ were performed using established protocols[61]. Briefly, lake water from three different depths of the chemocline was sampled directly from the CTD pump system into five 250 ml serum bottles per depth. Water was filled into the bottles from bottom to top, allowing 1–2 bottle volumes to overflow to minimize oxygen contamination before crimp-sealing the bottles headspace-free with butyl rubber stoppers. Back in the field laboratory, no more than 8 h after sampling, one bottle per depth was filtered onto pre-combusted (460 °C, 6 h) glass microfiber filters (GF/F, Whatman®, UK) for in situ natural abundance of C and N. $^{13}$C-labeled sodium bicarbonate (NaH$^{13}$CO$_3$, 98 atom% $^{13}$C, dissolved in autoclaved MilliQ water; Sigma-Aldrich) was injected (320 µL) into three bottles per depth, to achieve a final concentration of 160 µmol L$^{-1}$. Then, a volume of 5 ml $^{15}N_2$ gas (Cambridge Isotope Laboratories, >98 atom% $^{15}$N, Lot #: I-19197/AR0586172) was injected as a bubble into the same bottles and shaken for 20 min to equilibrate the $^{15}N_2$ gas. Sulfide solution was injected aiming for a final concentration of approximately 2 µM to remove trace oxygen contamination in the incubation bottles. Finally, the $^{15}N_2$ gas bubble was replaced with anoxic in situ lake water from the respective depth. The bottles, together with one untreated control bottle per depth (containing unamended lake water), were incubated for a full light-dark cycle (13 h light, 11 h dark) under natural light conditions (0–8267 Lux, average: 247 Lux, median: 10.8 Lux, as determined by a HOBO pendant data logger, Onset Computer Corporation, Bourne, USA) in a water bath kept at ~12 °C.

After incubation, samples were filtered onto pre-combusted GF/F filters. The filters were dried at room temperature and frozen at −20 °C for transport and storage. In addition, subsamples for nanoscale secondary ion mass spectrometry (nanoSIMS) analysis and for the determination of $^{13}$C and $^{15}$N enrichments in the substrate pools were taken from all bottles amended with $^{13}$C and $^{15}$N. NanoSIMS samples were fixed with 2% (final w/v) formaldehyde solution for 1 h at room temperature, prior to filtration onto gold-sputtered 0.22 µm polycarbonate membrane filters (GTTP Isopore™, Merck Millipore, USA). Subsamples for label% determinations were taken in gas-tight glass vials (Exetainer Labco, UK) and biological activity was terminated with HgCl$_2$.

Samples on GF/F filters were analyzed for C and N content and the respective isotopic composition by an elemental analyzer (Thermo Flash EA, 1112 Series) coupled to a continuous-flow isotope ratio mass spectrometer (Delta Plus XP IRMS; Thermo Finnigan, Dreieich, Germany). Enrichment of $^{15}$N in the $N_2$ pool was determined using a membrane inlet mass spectrometer (MIMS; GAM200, IPI). Enrichment of $^{13}$C in the dissolved inorganic carbon pool was determined from $^{13}$C/$^{12}$C-CO$_2$ ratios after sample acidification with phosphoric acid using cavity ring-down spectroscopy (G2201-I coupled to a Liaison A0301, Picarro Inc., connected to an AutoMate Prep Device, Bushnell, USA). In addition, we tested the used $^{15}N_2$ gas bottle for contamination with $^{15}$N-ammonia[62]. Briefly, a 2 ml subsample of the used $^{15}N_2$ gas was injected into a 12 ml gas-tight glass vial (Exetainer) filled with MilliQ (pH < 6). Any potentially present ammonia/ammonium was dissolved into the MilliQ via shaking and incubation overnight. Next, the liquid sample was repeatedly bubbled with helium to remove residual $^{15}N_2$ gas. 40 µmol L$^{-1}$ of $^{14}$N-ammonium (as (NH$_4$)$_2$SO$_4$) was added to the sample as a carrier for the gas chromatography isotope ratio mass spectrometry (GC-IRMS) analysis. Total dissolved ammonium was oxidized to $N_2$ using alkaline hypobromite iodine[63], which combines two ammonia/ammonium molecules to $N_2$. Due to the added $^{14}$N-ammonium, any $^{15}$N-ammonium contamination would result in the formation of $^{29}N_2$, thus enabling us to differentiate $^{15}$N-ammonium contamination from potential residues of the tested $^{30}N_2$ gas. After hypobromite conversion, $N_2$ isotopes were analyzed by GC-IRMS on a customized TraceGas coupled to a multi collector IsoPrime100. Together with the samples, we analyzed hypobromite treated $^{15}$N-ammonium standards[64]. We detected no $^{15}$N-ammonia contamination in the used $^{15}N_2$ bottle (detection limit: 5 pmol in 2 ml gas, equaling 0.056 ppm).

Bulk $N_2$ and $CO_2$ fixation rates were calculated from the incorporation of $^{15}$N and $^{13}$C, respectively, into biomass as described in Mohr et al.[65], according to Eq. 1

$$N_2 \text{ fixation rate} \left(nmol\ N\ L^{-1}d^{-1}\right) = \frac{(at\%PON_{sample} - at\%PON_{NA})}{(at\%N_2 - at\%PON_{NA})} * \frac{[PON]}{time} \qquad (Eq.1)$$

With at%PON$_{sample}$, at%PON$_{NA}$, and at%N$_2$ representing the atomic%$^{15}$N in the particulate organic nitrogen (PON) of the incubated sample, the incubated natural abundance sample and the $N_2$ pool, respectively. [PON] is the concentration of PON in the incubated sample per L, time is the incubation time in days. $CO_2$ fixation rates were calculated accordingly, using atomic%$^{13}$C, and particulate organic carbon (POC).

We further calculated the amount of N required to sustain the measured autotrophic C fixation, based on the depth-specific biomass C/N ratio (determined for the in situ bulk biomass in the respective water depth using an elemental analyzer), according to Eq. 2

$$Autotrophic\ N\ demand = \frac{CO_2\ fixation\ rate}{Depth\ specific\ N\ biomass\ C\ to\ N\ ratio} \qquad (Eq.2)$$

Calculations were performed using Microsoft Excel 2016.

**Re-analysis of metagenomes from 2014.** Raw metagenomic sequence data from the chemocline of Lake Cadagno (SAMEA4666021) and an enrichment culture from the lake (SAMEA4666022) previously published by Berg et al.[29], was downloaded from NCBI. Reads were adapter- and quality-trimmed using BBDuk[66] v37.24 (phred score 10, minimum length 50 bp). Reads from both metagenomes were co-assembled using metaSPAdes[67] version 3.14.0. Assembly statistics are shown in Table S6. Scaffolds were renamed to match the anvi'o workflow requirements, using anvi-script-reformat-fasta with anvi'o[68] version 6.2. Reads from both chemocline and enrichment metagenomes were mapped to the assembled and renamed scaffolds using Bowtie 2[69] version 2.3.5.1. Sorted and indexed bam files were created using samtools[70] 1.10 and anvi-init-bam. We followed the anvi'o metagenomics workflow, including gene prediction with Prodigal[71] V2.6.3, functional annotations with NCBI COGs and GhostKOALA/KEGG[72], and taxonomic gene annotation with Centrifuge[73] version 1.0.4. Predicted gene sequences annotated as either nifH, nifD, or nifK were manually validated by a blastx search[74] to the NCBI non-redundant protein sequences (nr) database. Only sequences whose best blastx hits matched their respective functional annotation were retained as nif genes.

A merged anvi'o profile database was created using sequence coverage information from both metagenomes and a minimum contig length of 1000 bp. External binning of scaffolds was performed using MetaBAT[75] version 2.12.1, using a minimum scaffold length of 1000 bp and minimum mean coverage of a scaffold in each library of 0. We additionally binned the scaffolds with concoct[76] 1.1.0 and used DAS Tool[77] version 1.1.2 to obtain an optimal, non-redundant set of bins that is based on the results of the two individual binning methods. The resulting bins were imported into anvi'o. A manual binning of as yet unbinned scaffolds was performed in anvio. Briefly, the unbinned fraction of the anvi'o database was visualized with anvi-refine, and several additional bins were compiled based on the anvi'o hierarchical-clustering. All bins containing at least one of the structural nitrogenase genes (MoFe nitrogenase: nifH, nifD, or nifK; VFe nitrogenase: vnfH, vnfD, vnfK, vnfG; FeFe nitrogenase: anfH, anfD, anfK, anfG) were identified, manually refined using anvi-refine and saved as an independent collection (in the following termed nif-MAGs). Nif-MAGs were further analyzed for completeness and contamination using CheckM[78] v1.0.18 and taxonomically classified using GTDB-Tk[79] v1.1.0.

**Metatranscriptome sequencing and analysis.** Samples for nucleic acid extraction (2018 campaign) were obtained by filtration of 100 ml lake water per incubation depth onto 0.22 µm GVWP Durapore® membrane filters (Merck Millipore, USA). Samples were immediately frozen and transported at -20 °C. The samples were then stored at −80 °C until further processing. Total RNA was extracted using the RNeasy PowerWater Kit (Qiagen, Germany) according to the manufacturer's protocol, including an on-column DNase digestion. Total RNA was sequenced (2x250bp) using Illumina HiSeq2500. Sequencing, including library preparation, was performed by the Max Planck-Genome-Centre Cologne (https://mpgc.mpipz.mpg.de/home/).

Raw reads were adapter- and quality-trimmed using BBDuk[66] v37.82 (phred score 10, minimum length 50 bp, tbo and tpe flag). Read-pairs were merged using BBmerge[80] v37.82. Mapping of the merged transcriptome reads to the metagenome assembly (see above) was performed using Bowtie 2[69] version 2.3.5.1 as described above. featureCounts[81] version v2.0.1 was used to extract read counts per predicted gene from the bam files.

To check for transcription of additional nifH genes and genes encoding for alternative nitrogenases that were not included in the metagenome assembly (and therefore not captured by the mapping approach), ROCker[82], and HMMER (hmmer.org) searches were performed. First, the merged reads were sorted into rRNA and mRNA datasets using SortMeRNA[83] version 4.2.0 (Table S4). ROCker searches were performed on the mRNA datasets using three custom nifH ROCker models (read lengths 250, 450, and 500, respectively). For hmmsearch, mRNA read sequences were translated into amino acid sequences using prodigal (meta mode) and searched with the TIGRFAMs models TIGR01287, TIGR01861, TIGR02929, TIGR02931, TIGR01860, TIGR02930, and TIGR02932 for nif, anf, and vnf genes using a bit score cutoff of 100. From all hits obtained through ROCker and HMMER, only reads that had not mapped against the metagenome assembly were

kept. The sequences were then manually validated as described above (blastx search).

The relative abundance of SSU rRNA transcripts was used as a combined proxy for cell abundance and cellular ribosome content (reflecting activity). We used phyloFlash[84] with the SILVA 138 SSU database and a read limit of 900,000 per sample to determine the taxonomic composition of the SSU rRNA reads. The results were visualized in R[85] using ggplot2[86].

**Nif amino acid trees**. A NifH amino acid sequence dataset was compiled that contained NifH sequences originating from the Lake Cadagno metagenome assembly and NifH sequences originating from the three metatranscriptomes. A blastp search[74] of the Cadagno NifH sequences to the NCBI nr database was performed to identify closely related, full-length NifH reference sequences from taxonomically classified organisms. Manually selected, representative reference sequences were downloaded from NCBI. A multiple amino acid sequence alignment of all sequences, including Cadagno and reference NifH sequences, was obtained with MAFFT[87] version 6.717b. RAxML[88] version 8.2.12 with the PROTGAMMAWAGF substitution model was used to calculate a maximum likelihood tree based on the alignment, but only including full-length NifH sequences. Partial NifH sequences which were excluded from initial tree calculation were then added to the tree using Taxtastic v0.9.0, pplacer, and guppy[89] v1.1. alpha19-0-g807f6f3.

In addition, we constructed NifD and NifK phylogenetic trees, including the NifD/NifK sequences retrieved from the MAGs. Reference sequences sharing >95% sequence identity to any of the MAG NifD/NifK sequences were identified with a blastp search[74] to the NCBI nr database. Multiple sequence alignments were obtained with MAFFT[87]. All full-length sequences were used to construct base trees with RAxML[88] and 100 bootstraps in ARB[90]. The ARB Parsimony function was employed to add partial sequences to the base trees.

The resulting trees were visualized in iTOL[91].

**FISH, cell counts, and cell sizes**. From each incubation depth, 10–30 ml lake water was filtered onto 0.22 μm polycarbonate membrane filters (GTTP Iso-pore™, Merck Millipore, USA). The filters were fixed in 2% formaldehyde solution in sterile-filtered lake water for 10–12 h at 4 °C and then washed with MilliQ water. The filters were frozen and stored at −20 °C until further processing.

The 16S rRNA FISH probe "Thiosyn459" (Table S7), exclusively targeting *T. syntrophicum* Cad16, was designed in ARB[90]. In addition, two competitor probes and four helper probes[92] were designed (Table S7) to ensure efficient and specific binding of the probe to the target. All FISH probes and respective formamide concentrations are listed in Table S7. Probes, but not helpers and competitors, were double-labeled with either Atto488 or Atto594 fluorophores. Samples were embedded in 0.05% low melting point agarose. Cells were permeabilized with lysozyme (1.5 mg ml$^{-1}$) for 30 min at 37 °C. Hybridization was performed for 2–4.5 h at 46 °C. Washing included 15 min in washing buffer at 48 °C and 20 min in 1× PBS buffer at room temperature. We used the hybridization and washing buffers described in Barrero-Canosa et al.[93] to reduce background fluorescence. Cells were counterstained with DAPI.

Samples were analyzed using a Zeiss Axio Imager.M2 microscope equipped with a Zeiss Axiocam 506 mono camera. Z-stack images were taken and the number of fluorescently labeled cells per image was counted for the individual probes. For each PSB population, we analyzed ≥38 randomly selected fields of view and ≥54 target cells, on one filter replicate each (see Supplementary File S1). Total cell counts were obtained in triplicates through flow cytometry as described in Danza et al.[94].

For cluster-forming organisms (*Thiodictyon syntrophicum*, *Lamprocystis purpurea*, *Lamprocystis roseopersicina*, and *Lamprocystis* spp.), the cell size (length and width, for biovolume and C-content calculations, see section below) of 100 cells per population was determined from the maximum-intensity projection of the z-stack images using the Zeiss Zen blue software 3.2.

**Single-cell analysis with nanoSIMS**. For nanoSIMS analyses, we chose the replicate sample from 13.7 m depth that exhibited the highest bulk N$_2$ fixation rate. Random spots were marked with a laser microdissection microscope (6000 B, Leica) on the gold-sputtered GTTP filter covered with cells incubated with $^{15}$N$_2$ and $^{13}$CO$_2$. After laser marking, FISH was performed as described above. For analysis of *Thiodictyon* cells, no permeabilization was performed, while for analysis of the other population's permeabilization was reduced to 15 min at 37 °C using 2 mg ml$^{-1}$ Lysozyme. Within one hybridization reaction, we simultaneously applied Apur453 with S453D and Laro453 with Cmok453, each probe double labeled with different fluorescent dyes (Atto488 and Atto594).

Single-cell $^{15}$N- and $^{13}$C-assimilation from incubation experiments with $^{15}$N$_2$ and $^{13}$CO$_2$ was measured using a nanoSIMS 50 L instrument (CAMECA), as described in Martínez-Pérez et al.[53]. Briefly, instrument precision was monitored regularly on graphite planchet. Samples were pre-sputtered with a Cs$^+$ beam (~300 pA) before the measurements with a beam current of around 1.5 pA. The diameter of the primary beam was tuned <100 nm. Measurements were carried out with a

dwelling time of 1 ms per pixel and a raster size of $6 \times 6$ to $48 \times 48$ μm. The pixel resolution for all measurements was $256 \times 256$. Between 20 and 70 planes were recorded for every measurement.

Measurements were analyzed using the Look@NanoSIMS software[95] version 2015-10-20 as described in Martínez-Pérez et al.[53]. Briefly, the recorded secondary ion images were drift corrected and accumulated. Using the corresponding epifluorescence microscopic images, regions of interest containing the target cells were defined. Ratios of $^{15}$N/($^{15}$N + $^{14}$N) and $^{13}$C/($^{13}$C + $^{12}$C) were used to calculate cell-specific N$_2$ and CO$_2$ fixation rate[96] only when the overall enrichment Poisson error across all planes of a given cell was <5% (using $^{15}$N/$^{14}$N and $^{13}$C/$^{12}$C ratios) according to Eq. 3

$$Cell\ specific\ N\ fixation\ rate(fmol\ N\ cell^{-1}d^{-1}) = \frac{at\%excess^{15}N_{cell}}{(at\%excess^{15}N_{medium})} * \frac{N_{cell}}{time}$$

(Eq.3)

Where at%excess$^{15}$N$_{cell}$ and at%excess $^{15}$N$_{medium}$ is the excess $^{15}$N atom percent enrichment above natural abundance measured for a given cell (from nanoSIMS measurements) and the incubation medium ($^{15}$N$_2$ added to lake water, from MIMS measurements), N$_{cell}$ is the amount of N per cell in fmol (see Eqs. 4 and 5 and text below) and time is incubation time in days. Cell-specific C fixation rates were calculated accordingly, using atomic%$^{13}$C, and cellular carbon content (C$_{cell}$, see Eqs. 4 and 5, and text below).

One *Lamprocystis* spp. cell that did not fix N$_2$ and CO$_2$ during the incubation is not shown in the figures (with the exception of Fig. S5) but was included for all calculations. For single-cell rate calculations, we did not account for isotope dilution effects, which have been reported for FISH-treated samples e.g.[97]. To assess a possible dilution effect from FISH on the isotopic composition ($^{15}$N/($^{15}$N + $^{14}$N) and $^{13}$C/($^{13}$C + $^{12}$C)) of PSB in our experiments, we performed nanoSIMS analysis of *C. okenii* cells that did not undergo FISH but were identified based on morphology. Using the same replicate sample as for all other nanoSIMS analyses (but untreated by FISH or DAPI staining), we found no significant difference in $^{15}$N enrichment of FISH-treated and untreated *C. okenii* cells, and slightly lower $^{13}$C enrichment in untreated compared to FISH-treated cells (Supplementary Fig. S9). The variability in $^{15}$N/($^{15}$N + $^{14}$N) ratios across measured cells was calculated following Svedén et al.[98] (Fig. S10), which confirmed that sufficient cells of all populations have been analyzed. The cellular carbon content of the comparably large PSB cells was estimated according to Verity et al.[99] according to Eq. 4

$$C_{cell}(fmol\ C\ cell^{-1}) = 0.433*BV^{0.863}*\frac{1000}{C_{molar\ mass}}$$

(Eq.4)

BV is the cellular biovolume in μm$^3$ and C$_{molar\ mass}$ is the molar mass of carbon (12 g mol$^{-1}$). Cellular nitrogen content (N$_{cell}$) was calculated based on the cellular carbon content (C$_{cell}$), and assuming a C:N ratio of 8.6:1, as determined for the in situ bulk biomass in the respective water depth using an elemental analyzer.

Biovolumes (BV) of PSB were calculated using the formula for a prolate sphere[100], according to Eq. 5

$$BV\ (\mu m^3) = \frac{\pi}{6} * width^2 * length$$

(Eq.5)

Cell width (i.e., cross-section), and length are given in μm.

Biovolumes of the non-cluster forming *C. okenii* cells were calculated from cell (region of interest) dimensions obtained from nanoSIMS data and used for the calculation of a cell-specific carbon content (C$_{cell}$). For the cluster-forming PSB, biovolumes of 100 cells per population were calculated using the cell dimensions determined from FISH images (see above). The median biovolume was used for the calculation of the population-specific cellular carbon content.

**Reporting summary**. Further information on research design is available in the Nature Research Reporting Summary linked to this article.

## Data availability

The sequence data generated in this study are deposited in the NCBI database. Metatranscriptomic data is deposited under BioProject number PRJNA693537 and BioSample numbers SAMN17390591, SAMN17390592, and SAMN17390593. *Nif*-gene encoding MAGs, generated from metagenome data retrieved from BioProject PRJEB22995, are deposited under BioProject PRJNA697932 and BioSample numbers SAMN17492688 to SAMN17492723. Cell counts, fixation rates and detection limits, NifH, NifD, and NifK tree sequences, and detailed MAG information are available in Supplementary File 1. The databases NCBI SRA, NCBI non-redundant protein sequences (nr), and SILVA 138 SSU were used for data analyses. Source data are provided with this paper.

## Code availability

The custom R function used for the correction of sampling depth is accessible via https://github.com/mirimarine/N2-fixation-in-Lake-Cadagno.

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

## Acknowledgements
We thank G. Klockgether, K. Imhoff, G. Lavik, B. Fuchs, D. Tienken, S. Lilienthal, N. Rujanski, J. Berlinghof, S. Roman, C. Stengel, and S. Piosek for technical assistance; M. Tonolla and S. Roman for logistics; and J. Klatt, P. F. Hach, P. Pjevac, and J. Dürschlag for fruitful discussions. We are grateful to the Alpine Biology Center Foundation (Switzerland) for use of its research facilities. The research was funded by the Max Planck Society.

## Author contributions
M.M.M.K., W.M., K.K. and M.P. designed the study. M.P., J.S.B., C.S., N.S. and M.M. M.K. performed fieldwork. L.H.E.W. provided molybdenum measurements. N.S. conducted flow cytometry analyses. M.P. conducted FISH analyses and analyzed process rates with W.M. Molecular analyses were carried out by M.P., B.T. and H.K.M. NanoSIMS analyses were done by S.L., A.T.K. and K.K. Flux calculations were done by J.S.B. The paper was written by M.P., K.K., M.M.M.K. and J.S.B. with contributions from all co-authors.

## Funding

## Competing interests
The authors declare no competing interests.
