## [Peer Review File · Nature Communications]

REVIEWER COMMENTS

Reviewer #1 (Remarks to the Author):

Review of Purple sulfur bacteria fix N₂ via molybdenum-nitrogenase in a low molybdenum Proterozoic ocean analogue

The current study investigates the N₂ fixation capability of purple sulfur bacteria (PSB) in an analogue system to the sulfidic Proterozoic continental margins. Using a combination of meta-omics analyses combined with visualization techniques and stable isotope labeling the authors brings evidence that PSB are able to fix N₂ at low molybdenum concentrations using exclusively molybdenum-nitrogenase and conclude that PSB may have been responsible for N₂ fixation in the Proterozoic ocean, in contrast to previous believes.

The study idea and interpretation of the results are novel, well presented and deserves consideration for publication. The study concept is very interesting and brings new aspects to the nitrogen fixation processes and the PSB microorganisms able to transform N₂ gas into bioavailable nitrogen and their potential role as drivers of N process.

The diversity of complementary techniques used to answer the study questions is well selected and the results and data interpretation are excellently combined leading to a clear outcome which challenges previous views with implications for early life.

I have, however a number of concerns, mostly related to the number of single cell analyzed used further for rate calculations, detailed bellow.

General comments:

The nitrogen fixation by green sulfur bacterium was previously reported in this lake (Zimmermann et al., 2015), as the authors specify in the supplementary file. However the authors did not discussed the findings of this study, particularly on the fact that N₂ fixation was shown to occur in the absence of ammonium only, which is in contrast with observations made here.

Moreover, the authors shows that N₂ fixation seems to provide just a part of the nitrogen demands of the PSB cells, the rest being covered by other N sources. This aspect is not discussed in the context of Proterozoic ocean. Do the authors have a hypothesis, in the light of their findings, on how the PSB in Proterozoic Ocean meet their N demands and fuel others based exclusively on N₂ fixation? This comment is in relation to specific comments lines 349-353 bellow

Specific Comments

Line 555: It is not clear if the nanoSIMS measurements were done from all triplicates or just from one bottle. If only from one labeling experiment, can the authors specify why cells from duplicate experiments were no analyzed? As the single cell measurements are the base for the N₂- fixation rate calculations it is important a higher number of cells to be analyzed. For example the rate calculations for *C. okenii* is based on a very small number of cells (35 cells) and for *Lamprocystis* groups up to 56 cells. Particularly for *Lamprocystis* groups that prefer to grow in aggregates which are easily spotted under fluorescence microscope and mapped for nanoSIMS would have been not a huge time investment to increase the number of analyzed cells.

Along the same lines the ¹⁵N and ¹³C uptake and C and N fixation rates between individual cells of the same group is pretty large e.g. *C. okenii*, e.g. ¹⁵N uptake ranging from natural abundance values to about 3 at% (Fig S3), N₂ fixation rates from 0.40 to 10 fmol N/cell/day; CO₂ fixation rates from 20 to 400 fmolC/cell/day, yet these are the bases for population-specific rates calculated using the average cell abundance and average single cell N₂ fixation rate. The statistics of these values will have improved significantly if more single cells would have been analyzed.

I suggest to increase the number of single cells analyzed, particularly for *C. okenii* and *Lamprocystis*

groups or to specify that the N and C fixation rates and the interpretation based on this should be treated with caution due to a relatively low cell numbers analyzed.

Lines 548-554; 567-571: The authors chose not to account for isotope dilution in their calculations although previous studies (Musat et al., 2014; Meyer et al., 2020) showed that hybridization events, particularly when permeabilization is employed will lead to tremendous isotope dilution, particularly of the ^{15}N . In the present study the authors besides the permeabilization step also employed dual hybridization events which increase the potential of higher isotope dilution. Authors justified the missing calculations on the isotopic effects by the fact that the combined activity of the PSB groups as resulted from single cell analysis "almost" match the bulk fixation activity. Can the authors be more precise? For contribution to C fixation seems just a small portion of total bulk is covered by the individual groups (Fig. S2).

Lines 255-256: is there a std for the cell counts at different depths?

Lines 349-350: It is clear that the anoxygenic phototrophs are able to fix N_2 in the presence of ammonium. However it is not clear how such organisms may have survived protozoic ocean conditions as obviously the amount of N fixed by these do not meet the cellular demand.

Lines 352-353: how? since it was shown by co-authors that the cells needs also an alternative source of N to meet their N demands?

Lines 539-541 and supplementary excel file: Information is missing on the FISH cell counting. How many independent field of views were used for counting? How many replicate filters or samples. Duplicate experiments? Are the values in the excel files average values?

Reviewer #2 (Remarks to the Author):

The authors present new geochemical and molecular data from Lake Cadagno, demonstrating the presence of N_2 -fixing purple sulfur bacteria in a Proterozoic analogue environment. The results demonstrate that anoxygenic phototrophs can act as the major diazotrophs in anoxic ecosystems and that the MoFe nitrogenase enzyme is not inhibited by low Mo concentrations in the water column. These findings would imply that the early Precambrian oceans were perhaps more habitable than commonly assumed, because nitrogen may not have been as difficult to acquire as previously proposed.

The study is very elegant and addresses a fundamental aspect of habitability and microbial ecology on the early Earth. The results are compelling and might represent a significant advance in our understanding of how nitrogen could be acquired in anoxic ecosystems with low supplies of molybdenum. I would therefore expect this paper to be very impactful and widely cited.

My main comment is that the authors should explain better how this study differs from previous work on Lake Cadagno where N_2 fixation was demonstrated to occur in green sulfur bacteria in the same environment (Halm et al. 2009). The thrust here is on purple sulfur bacteria, but it should be explained better how PSB compare in ecological importance to GSB and how their ecological role may project back in time to Proterozoic conditions.

I stress that the molecular biological analyses are outside of my area of expertise. I therefore only have a few additional comments that the authors may want to address:

I. 36: Change to 'was one of the first'. There would also have been NH_4^+ generated in hydrothermal processes (e.g., Brandes et al. 1998).

II. 82-84: It is not explained well throughout the paper how the ecological importance of PSB compare to that of GSB. As stated here, the ability of GSB to fix N_2 in the environment has previously been demonstrated, and PSB were previously known to possess the genetic toolkit for diazotrophy. It should therefore be explained better to the reader how this study is different and why it represents an

important advance.

I. 91: Specify how low the Mo concentrations are in units of nM and remind the reader of the concentration found seawater (105 nM). That would help illustrate how low these levels really are.

II. 131-137: I agree with this interpretation, but it would be important to rule out that this lake receives significant input of riverine input, which could contribute particulate organic N with a composition near 0 permil.

II. 155-158: It is not clear to me what is meant by autotrophic N demand. Is this the N demand of autotrophic (CO₂-fixing) organisms, or is it the N demand of N-fixing organisms? How was the demand calculated? Please clarify.

I. 161: How was this ammonium flux calculated? Is this derived from the measured concentration profiles and assuming upward diffusion?

Best wishes,
Eva Stüeken

Reviewer #3 (Remarks to the Author):

It has been proposed that the emergence of nitrogen-fixing organisms has enabled the diversification of life on Earth. The environmental conditions on early Earth are believed to be determining factors for these evolutionary landmark events. Therefore the study of nitrogen fixation in environments that mimic these ancient settings can provide insight into the primitive biochemical catalysts. This report has explored environmental microbial samples extracted from Lake Cadagno. Specifically, the samples characterized in this study were obtained from 14 m depth at transient conditions for decreasing Mo and increasing S₂- concentrations. The report shows that, under these conditions, purple sulfur bacteria can actively fix nitrogen. This finding, combined metagenomic and metatranscriptomic analyses, demonstrates that *Chromatium okenii* is the dominant species responsible for this biological activity. The report is relevant as it provides experimental evidence for N₂ fixation in environmental samples and expands the taxonomic and environmental diversity of biological nitrogen fixation. The experimental approach is sound, and conclusions are well-supported by the experimental evidence provided in this report.

However, the findings described in this study are not considered ground-breaking as a draft of the genome of *C. okenii* was previously reported and the presence of nitrogen fixation genes were noted. Perhaps, the report could be strengthened with the expansion of phylogenetic analysis of *nif* genes to demonstrate that the *NifDK* represents a more ancient copy of nitrogenase and the *nif* accessory genes enable nitrogen fixation under environmental conditions. Based on this assessment, I believe this contribution as presented is borderline for *Nat. Comm.*, and perhaps would be a more suitable report to a specialized journal.

Below I list selected points that the author may want to consider when revising this report.

1_ Other studied diazotrophs have been shown to utilize Mo-dependent nitrogenase at low nM concentrations of Mo, concentrations of which are similar to those determined in this study (Fig 1). Therefore, the claim that is an unexpected finding is not well-supported by previous literature.

2_ I believe the authors have missed out on the opportunity to report the presence of additional *nif* genes in the *C. okenii* genome. This would strengthen the proposal for this species to be a diazotroph capable of fixing nitrogen under low Mo concentration. For instance, this organism contains additional genes that have been shown to be essential for nitrogen fixation (e.g. *nifEN*, *nifB*). Interestingly, a copy of *nifQ* is found immediately upstream of a *nifH* gene, supporting the idea that this organism has a system to operate under low Mo concentrations.

3_ Sequence analysis of *NifD* and *NifK* sequences also suggests that they are phylogenetic distinct from well-studied *NifDK* sequences and perhaps represent an evolutionary representative of ancient

nitrogen fixation.

4_ Overall Mo-dependent nitrogenase is known to be more predominant and alternative enzymes coded by *vnf* and *anf* systems. The results presented in this study are compatible with the model that Mo-independent systems have emerged later on Earth history.

Response to Referees

Purple sulfur bacteria fix N₂ via molybdenum-nitrogenase in a low molybdenum Proterozoic ocean analogue, Philippi et al.

Reviewer #1 (Remarks to the Author):

Review of Purple sulfur bacteria fix N₂ via molybdenum-nitrogenase in a low molybdenum Proterozoic ocean analogue

The current study investigates the N₂ fixation capability of purple sulfur bacteria (PSB) in an analogue system to the sulfidic Proterozoic continental margins. Using a combination of meta-omics analyses combined with visualization techniques and stable isotope labeling the authors brings evidence that PSB are able to fix N₂ at low molybdenum concentrations using exclusively molybdenum-nitrogenase and conclude that PSB may have been responsible for N₂ fixation in the Proterozoic ocean, in contrast to previous believes.

The study idea and interpretation of the results are novel, well presented and deserves consideration for publication. The study concept is very interesting and brings new aspects to the nitrogen fixation processes and the PSB microorganisms able to transform N₂ gas into bioavailable nitrogen and their potential role as drivers of N process.

The diversity of complementary techniques used to answer the study questions is well selected and the results and data interpretation are excellently combined leading to a clear outcome which challenges previous views with implications for early life.

I have, however a number of concerns, mostly related to the number of single cell analyzed used further for rate calculations, detailed bellow.

We thank the reviewer for their positive feedback and constructive comments. We have obtained and included additional data and addressed their comments in the revised manuscript, as detailed below.

General comments:

The nitrogen fixation by green sulfur bacterium was previously reported in this lake (Zimmermann et al., 2015), as the authors specify in the supplementary file. However the authors did not discussed the findings of this study, particularly on the fact that N₂ fixation was shown to occur in the absence of ammonium only, which is in contrast with observations made here.

- 1) Halm et al (2009, EMI) and Zimmermann et al (2015, Frontiers Microbiol) show that bulk in situ N₂ fixation in Lake Cadagno occurs even in presence of high ammonium

concentrations (up to 1000 $\mu\text{M NH}_4^+$), which is in line with our findings. At the same time, Zimmermann et al determined the effect of ammonium on single cell N_2 fixation activity by the GSB *Chlorobium phaeobacteroides* using nanoSIMS and showed that this organism did not fix N_2 in presence of ammonium. They therefore attributed the measured bulk N_2 fixation activity to diazotrophs other than the GSB *Chlorobium phaeobacteroides*. Our data suggests that the N_2 -fixing PSB could be the organisms responsible for the unaccounted-for bulk N_2 fixation activity in these previous studies. We now discuss our findings in relation to these studies in L. 159-163 and L. 347-350.

Moreover, the authors shows that N_2 fixation seems to provide just a part of the nitrogen demands of the PSB cells, the rest being covered by other N sources. This aspect is not discussed in the context of Proterozoic ocean. Do the authors have a hypothesis, in the light of their findings, on how the PSB in Proterozoic Ocean meet their N demands and fuel others based exclusively on N_2 fixation? This comment is in relation to specific comments lines 349-353 bellow

- 2) Although the PSB single cell N_2 fixation rates we measured in Lake Cadagno were high, these rates may be even higher under conditions of more severe bioavailable N limitation, as likely was the case in the Proterozoic ocean. In our incubations, ammonium was present and thus, PSB probably did not reach their maximum N_2 -fixing capacity. A strong dependence of N_2 fixation activity on ammonium concentrations has previously been shown for diazotrophs, e.g. *Klebsiella*, where N_2 fixation increased with decreasing ammonium (Schreiber, 2016, Nature Microbiology). Therefore, we suggest that in the Proterozoic ocean, where bioavailable N was severely limiting, PSB derived all/most of their cellular nitrogen demand from N_2 fixation. We now address this point in L. 334-335 and 399-404.

Specific Comments

Line 555: It is not clear if the nanoSIMS measurements were done from all triplicates or just from one bottle. If only from one labeling experiment, can the authors specify why cells from duplicate experiments were no analyzed?

- 3) The nanoSIMS analyses presented were all performed on cells from one replicate (now specified in L. 310-311 and 614-615, and in the Supplementary file S1 in the sheet "Fixation_rates"), due to the time and cost intensive nanoSIMS measurements. We have now performed nanoSIMS measurements on a substantial number of additional cells, nearly doubling the number of analyzed cells per population. We have now also included data from an additional independent biological replicate and the results confirm the high contribution of *C. okenii* to bulk N_2 fixation rates (see Supplementary Table S5 and Supplementary File S1), and revised manuscript L. 344-347.

As the single cell measurements are the base for the N_2 - fixation rate calculations it is important a higher number of cells to be analyzed. For example the rate calculations for *C. okenii* is based

on a very small number of cells (35 cells) and for Lamprocystis groups up to 56 cells. Particularly for Lamprocystis groups that prefer to grow in aggregates which are easily spotted under fluorescence microscope and mapped for nanoSIMS would have been not a huge time investment to increase the number of analyzed cells.

Along the same lines the ^{15}N and ^{13}C uptake and C and N fixation rates between individual cells of the same group is pretty large e.g. *C. okenii*, e.g. ^{15}N uptake ranging from natural abundance values to about 3 at% (Fig S3), N_2 fixation rates from 0.40 to 10 fmol N/cell/day; CO_2 fixation rates from 20 to 400 fmolC/cell/day, yet these are the bases for population-specific rates calculated using the average cell abundance and average single cell N_2 fixation rate. The statistics of these values will have improved significantly if more single cells would have been analyzed.

I suggest to increase the number of single cells analyzed, particularly for *C. okenii* and Lamprocystis groups or to specify that the N and C fixation rates and the interpretation based on this should be treated with caution due to a relatively low cell numbers analyzed.

- 4) We have now included a substantial number of additional single cell measurements for all *Lamprocystis* and *C. okenii* populations, nearly doubling the number of analyzed cells (≥ 79 cells) as requested and updated all figures and calculations accordingly. These newly calculated average cellular rates do not substantially differ from the rates reported in the previous version of the manuscript. We have now measured a large number of cells, considerably exceeding the number required to make statistically robust conclusions (calculation according to Svedén et al 2015, FEMS; data shown in former Fig. S7, now Fig. S10, see below).

Comparison of the previous and the current single cell data. N_2 fixation rate refers to the population average and is shown in fmol N cell $^{-1}$ d $^{-1}$ with the respective standard error. The contribution to bulk rate refers to the population's contribution to the average bulk N_2 fixation rate and is shown with the respective standard error.

Population	Previous data			Current data		
	Number of cells	N_2 fixation rate	Contribution to bulk rate (%)	Number of cells	N_2 fixation rate	Contribution to bulk rate (%)
C. okenii	35	3.22 ± 0.46	81.32 ± 14.07	79	3.29 ± 0.25	83.13 ± 10.24
L. sp.	52	0.67 ± 0.05	6.43 ± 1.22	130	0.79 ± 0.03	7.63 ± 1.38
L. roseop.	56	0.63 ± 0.02	7.13 ± 1.45	93	0.65 ± 0.03	7.35 ± 1.51
L. purpurea	45	0.62 ± 0.03	9.14 ± 1.51	116	0.69 ± 0.03	10.19 ± 1.66

Current Figure S10. The variability of ^{15}N at% (a-e) and ^{13}C at% (f-j) across measured cells. Black lines represent the total mean with the orange shading indicating $\leq 10\%$ deviation from the mean. The measured cells were ordered randomly and the mean enrichment was calculated with increasing number of regarded cells (white dots). The error bars represent the standard error of the mean enrichment. The plots were created in accordance with Svedén et al. FEMS.

Lines 548-554; 567-571: The authors chose not to account for isotope dilution in their calculations although previous studies (Musat et al., 2014; Meyer et al., 2020) showed that hybridization events, particularly when permeabilization is employed will lead to tremendous isotope dilution, particularly of the ^{15}N . In the present study the authors besides the permeabilization step also employed dual hybridization events which increase the potential of higher isotope dilution.

- 5) We agree that the fixation and FISH procedures can lead to isotope dilution and thus an underestimation of the actual single cell activity, as pointed out in the manuscript in L. 637-638. This dilution effect is dependent on growth stage, cell activity and type, and consequently, not all cells are equally affected. It is therefore difficult to accurately correct for an isotopic dilution effect in environmental samples. Nevertheless, we experimentally determined the impact of isotopic dilution on our nanoSIMS measurements by measuring N_2 and CO_2 uptake of untreated (no FISH, no DAPI staining) *C. okenii* cells (identified based on morphology) from the same replicate as used for all other nanoSIMS analyses. These untreated *C. okenii* cells were similarly enriched in ^{15}N as the *C. okenii* cells after FISH, with no statistically significant difference. For ^{13}C , we did observe a slight difference, surprisingly with slightly lower ^{13}C enrichment in untreated cells (1.59 and 1.45 mean ^{13}C -at% for FISH-treated and untreated cells, respectively; two sided, two-sample Wilcoxon Test, $W = 1463.5$, $p\text{-value} = 5.9 \times 10^{-5}$). This new data has been added as Supplementary Figure S9 and is described in L. 638-644.

We did not conduct dual-hybridizations but rather used two differently labeled probes in a simultaneous hybridization. We apologize for the confusion and have now clarified this in the Materials and Methods section of the revised manuscript (L. 620-622).

Authors justified the missing calculations on the isotopic effects by the fact that the combined activity of the PSB groups as resulted from single cell analysis “almost” match the bulk fixation activity. Can the authors be more precise? For contribution to C fixation seems just a small portion of total bulk is covered by the individual groups (Fig. S2).

- 6) Please see our answer (5) above regarding the potential dilution effect of FISH. We agree that the wording here was imprecise and have now specified that for the replicate where nanoSIMS was conducted, PSB explain $99.8 \pm 11.9\%$ of the bulk N_2 fixation activity. We have added this to the manuscript at L. 350-352.

Regarding the low proportion of bulk C-fixation explained by the individual PSB groups: the relatively smaller contribution of PSB to bulk CO_2 fixation compared to N_2 fixation is not unexpected, as large-celled algae were abundant in the analyzed water depth at the time of sampling (see microscopy image below). It was previously shown that phototrophic algae and anoxygenic phototrophs can contribute equally to CO_2 fixation in the upper Lake Cadagno chemocline (Camacho, 2001, Aquatic Sciences). This is in line with our findings (PSB contributed ca. 45% to bulk CO_2 fixation in the analyzed replicate).

We confirmed ^{13}C fixation activity of algae by nanoSIMS analysis. Based on cell size and ^{13}C -enrichment, an algal single cell CO_2 fixation rate of $\sim 370 \text{ fmol C day}^{-1}$ was determined. This is > 3 times higher than the average single cell CO_2 fixation rate of *C. okenii* ($116 \text{ fmol C cell}^{-1} \text{ day}^{-1}$).

Representative microscopy image of the sample analyzed with nanoSIMS. The sample was hybridized with a probe targeting *C. okenii* (red). Large-celled algae (indicated by white arrows) exhibited high levels of autofluorescence in red and green.

Lines 255-256: is there a std for the cell counts at different depths?

- 7) We have now updated the total cell counts measured with flow cytometry to average values with standard deviation of the three measured replicates in both the main text (L. 284-286) and the Supplementary file S1 (sheet "cell_counts").

Lines 349-350: It is clear that the anoxygenic phototrophs are able to fix N_2 in the presence of ammonium. However it is not clear how such organisms may have survived protozoic ocean conditions as obviously the amount of N fixed by these do not meet the cellular demand.

- 8) Please refer to our answer (2). We think that PSB, like other diazotrophs, increase their N₂ fixation activity when bioavailable N is depleted (i.e. meet a larger fraction or even all of their N demand by N₂ fixation; now incorporated in L. 334-335 and 399-404).

Lines 352-353: how? since it was shown by co-authors that the cells needs also an alternative source of N to meet their N demands?

- 9) Please refer to our answer (2) and (8).

Lines 539-541 and supplementary excel file: Information is missing on the FISH cell counting. How many independent field of views were used for counting? How many replicate filters or samples. Duplicate experiments? Are the values in the excel files average values?

- 10) We now specified how many fields of view per PSB population were counted, and from how many counted PSB cells we extrapolated to cell numbers per ml in the supplementary File S1 (cell_counts). We obtained cell counts from ≥ 38 randomly selected fields of view and ≥ 54 counted cells per population (now clarified in L. 604-606). We report average counts, which were obtained from one filter replicate (also L. 604-606).

Reviewer #2 (Remarks to the Author):

The authors present new geochemical and molecular data from Lake Cadagno, demonstrating the presence of N₂-fixing purple sulfur bacteria in a Proterozoic analogue environment. The results demonstrate that anoxygenic phototrophs can act as the major diazotrophs in anoxic ecosystems and that the MoFe nitrogenase enzyme is not inhibited by low Mo concentrations in the water column. These findings would imply that the early Precambrian oceans were perhaps more habitable than commonly assumed, because nitrogen may not have been as difficult to acquire as previously proposed.

The study is very elegant and addresses a fundamental aspect of habitability and microbial ecology on the early Earth. The results are compelling and might represent a significant advance in our understanding of how nitrogen could be acquired in anoxic ecosystems with low supplies of molybdenum. I would therefore expect this paper to be very impactful and widely cited.

We thank the reviewer for this encouraging and positive feedback.

1. My main comment is that the authors should explain better how this study differs from previous work on Lake Cadagno where N₂ fixation was demonstrated to occur in green sulfur bacteria in the same environment (Halm et al. 2009). The thrust here is on purple sulfur bacteria, but it should be explained better how PSB compare in ecological importance to GSB and how their ecological role may project back in time to Proterozoic conditions.

11) In Lake Cadagno, N₂ fixation was previously only investigated at a bulk level or at the single cell level for the GSB *Chlorobium phaeobacteroides* (Halm et al 2009 EMI; Zimmermann et al 2015 Frontiers Microbiol). Based on a few single cell measurements (4 cells in total), Halm et al hypothesized that GSB might be the main N₂ fixers in the Lake Cadagno chemocline, however, Zimmermann et al found that that GSB could not account for all of the measured bulk N₂ fixation activity and suggested that other diazotrophs might play an important role.

PSB were not investigated in either study, as at that time, there was no evidence that Lake Cadagno PSB had the genetic potential to fix N₂. In contrast to the previously used clone library approaches, our meta-omics data showed that PSB in Cadagno did have the potential to fix N₂, therefore we decided to investigate PSB N₂ fixation. Indeed, PSB might have contributed to N₂ fixation in these earlier studies, too.

We now emphasize how our study differs from previous studies on N₂ fixation in Lake Cadagno in L. 159-163 and 347-350.

To address how PSB compare to GSB in ecological importance, we have added information about the ecophysiological differences between these groups (different light and oxygen adaptations, growth rates and cellular activity) in L. 390-398.

I stress that the molecular biological analyses are outside of my area of expertise. I therefore only have a few additional comments that the authors may want to address:

2. I. 36: Change to 'was one of the first'. There would also have been NH₄⁺ generated in hydrothermal processes (e.g., Brandes et al. 1998).

12) Changed as suggested (L. 36).

3. II. 82-84: It is not explained well throughout the paper how the ecological importance of PSB compare to that of GSB. As stated here, the ability of GSB to fix N₂ in the environment has previously been demonstrated, and PSB were previously known to possess the genetic toolkit for diazotrophy. It should therefore be explained better to the reader how this study is different and why it represents an important advance.

13) Please also refer to our answer (11) regarding the differences between PSB and GSB. Briefly, although GSB N₂ fixation has been shown before, this occurred at low cellular rates (much lower than those we measured for PSB). We think that also in previous studies, PSB may have contributed substantially to Lake Cadagno N₂ fixation, however, PSB were not targeted in these earlier studies. We now address how our study differs from previous studies on N₂ fixation in Lake Cadagno in the manuscript at L. 159-163 and 347-350.

Indeed, the genetic potential for N₂ fixation has been shown before for PSB and this is an indication that an organism may fix N₂. Yet, their activity in situ, in natural communities has not been demonstrated before our study. This is however essential, as genetic

capacity does not automatically translate to activity in the environment. For example, *Sagittula castanea* has the genetic potential to fix N₂, and fixes N₂ in culture, but has so far not been found to fix N₂ in the environment (even though the investigated samples showed significant bulk N₂ fixation rates, Martinez-Perez et al. 2018 EMI).

We now highlight to the readers the necessity for in situ activity measurements to prove N₂ fixation by potential diazotrophs in L. 91-93.

4. I. 91: Specify how low the Mo concentrations are in units of nM and remind the reader of the concentration found seawater (105 nM). That would help illustrate how low these levels really are.

14) Changed as suggested (L. 98-100).

5. II. 131-137: I agree with this interpretation, but it would be important to rule out that this lake receives significant input of riverine input, which could contribute particulate organic N with a composition near 0 permil.

15) The anoxic bottom waters, in which we measured the low $\delta^{15}\text{N}$, are fed by underwater springs, while the oxygenated surface waters are supplied with meso- to oligotrophic waters from Lake Stabbio through a small stream and from surface-runoffs (Tonolla et al., 2017, Springer International Publishing). Indeed, we cannot fully exclude an external source of reduced N with a low $\delta^{15}\text{N}$ signature and have now included this caveat in the main text (L. 140-142). Our bulk ¹⁵N-N₂ incubation experiments however provide direct evidence for N₂ fixation being a major source of N in Lake Cadagno.

6. II. 155-158: It is not clear to me what is meant by autotrophic N demand. Is this the N demand of autotrophic (CO₂-fixing) organisms, or is it the N demand of N-fixing organisms? How was the demand calculated? Please clarify.

16) Autotrophic N-demand is the amount of N required to sustain the measured inorganic C-fixation (assuming a depth-specific biomass C:N ratio, which we determined from in situ biomass samples using an elemental analyzer). We have now clarified this in the manuscript (L. 167-169) and added the equation used for this calculation to the method section (L. 495-499). In addition, we added the respective numbers (depth-specific C:N ratios, depth-specific autotrophic N demand) and calculations to Supplementary file S1 (Sheet "Fixation_rates").

7. I. 161: How was this ammonium flux calculated? Is this derived from the measured concentration profiles and assuming upward diffusion?

17) Indeed, we calculated the ammonium flux from the measured concentration profile (i.e. upward diffusing ammonium) and assumed constant mixing in the chemocline. In addition to the methods (L. 448-454), we have also added a brief explanation to the main text (L. 176-179).

Best wishes,

Eva Stüeken

Reviewer #3 (Remarks to the Author):

It has been proposed that the emergence of nitrogen-fixing organisms has enabled the diversification of life on Earth. The environmental conditions on early Earth are believed to be determining factors for these evolutionary landmark events. Therefore the study of nitrogen fixation in environments that mimic these ancient settings can provide insight into the primitive biochemical catalysts. This report has explored environmental microbial samples extracted from Lake Cadagno. Specifically, the samples characterized in this study were obtained from 14 m depth at transient conditions for decreasing Mo and increasing S²⁻ concentrations. The report shows that, under these conditions, purple sulfur bacteria can actively fix nitrogen. This finding, combined metagenomic and metatranscriptomic analyses, demonstrates that *Chromatium okenii* is the dominant species responsible for this biological activity. The report is relevant as it provides experimental evidence for N₂ fixation in environmental samples and expands the taxonomic and environmental diversity of biological nitrogen fixation. The experimental approach is sound, and conclusions are well-supported by the experimental evidence provided in this report.

We thank the reviewer for their positive feedback.

However, the findings described in this study are not considered ground-breaking as a draft of the genome of *C. okenii* was previously reported and the presence of nitrogen fixation genes were noted.

18) We respectfully disagree, gene presence alone does not prove that a process is active and environmentally significant. Therefore, direct activity measurements under in situ conditions are vital to confirm an organism's activity. For example: The marine alphaproteobacterial diazotroph *Sagittula castanea* was shown to encode a full set of *nif* genes and was found to actively fix N₂ in pure culture (Martínez-Pérez et al., 2018, EMI). Yet, in situ analyses showed no detectable N₂ fixation activity by this organism in environmental samples that had significant bulk N₂ fixation rates (Martínez-Pérez et al., 2018, EMI). An exclusively sequence-based study might have erroneously attributed the significant environmental N₂ fixation rates in the Eastern Tropical South Pacific to *S. castanea*, which highlights the need for in situ observations to confirm activity. We have now emphasized these points in our manuscript (L. 92-93).

Perhaps, the report could be strengthened with the expansion of phylogenetic analysis of *nif* genes to demonstrate that the NifDK represents a more ancient copy of nitrogenase and the *nif* accessory genes enable nitrogen fixation under environmental conditions. Based on this assessment, I believe this contribution as presented is borderline for Nat. Comm., and perhaps would be a more suitable report to a specialized journal.

19) As suggested, we have performed the additional phylogenetic analyses of NifD and NifK (Supplementary Figures S3 and S4) to address a possible deep branching of *C. okenii nifD/nifK* genes. We did not observe the suggested distinct clustering of *Chromatium* NifD/K sequences. *Chromatium* sequences clustered together with the other PSB (also described for *nifD* of *C. okenii* in Luedin et al. 2019, Scientific reports), with no indication for deep branching/an ancient origin of PSB *nif* genes. This is in line with the fact that PSB most likely evolved after N₂ fixation itself. MoFe nitrogenase is believed to have evolved in a methanogen (Boyd, 2011, Geobiology), at least 3.2 Ga ago (Stüeken, 2015, Nature), and the earliest fossil PSB evidence dates back to 1.6 Ga ago (Brocks, 2005, Nature).

Below I list selected points that the author may want to consider when revising this report.

1_ Other studied diazotrophs have been shown to utilize Mo-dependent nitrogenase at low nM concentrations of Mo, concentrations of which are similar to those determined in this study (Fig 1). Therefore, the claim that is an unexpected finding is not well-supported by previous literature.

20) Indeed, it has been shown that Mo-dependent nitrogenase can be active under low Mo conditions (Zerkle et al., 2006, Geobiology; Glass et al., 2010, Limnol. Oceanogr.), as pointed out in the manuscript (L. 370-373). We have now further clarified this in L. 59-61. However, these studies were performed using cyanobacterial cultures which are normally growing in oxic environments that are not limited by the availability of Mo. Therefore, it was until now unclear if these cyanobacteria culture-based findings could directly be transferred to environmental organisms that generally live in Mo limiting anoxic environments, such as PSB. We show for the first time that microorganisms generally growing under Mo limitation actively fix N₂ using MoFe nitrogenase.

It is still widely assumed that the main function of alternative nitrogenases in environmental organisms is to take over N₂ fixation activity once Mo is limiting (Harwood 2020, Annual Review of Microbiology). Therefore, the intriguing lack of alternative nitrogenases in the genomes of PSB living permanently under Mo limiting conditions could have also suggested that these organisms fix only little to no N₂ and that their *nif* genes are hardly used. The high cellular N₂ fixation rates exhibited by the PSB in Lake Cadagno therefore are unexpected and novel.

2_ I believe the authors have missed out on the opportunity to report the presence of additional *nif* genes in the *C. okenii* genome. This would strengthen the proposal for this species to be a

diazotroph capable of fixing nitrogen under low Mo concentration. For instance, this organism contains additional genes that have been shown to be essential for nitrogen fixation (e.g. *nifEN*, *nifB*). Interestingly, a copy of *nifQ* is found immediately upstream of a *nifH* gene, supporting the idea that this organism has a system to operate under low Mo concentrations.

21) As suggested, we have now included a more detailed discussion of the detected *nif* genes in the Lake Cadagno *C. okenii* population in Supplementary Note S1 (Supplement L. 20-36). Indeed, in line with the measured single cell N₂ fixation activity in situ, the Lake Cadagno *C. okenii* population encodes for almost all genes considered essential for N₂ fixation, including *nifE*, *nifB* and *nifQ* (Dos Santos et al. 2012, BMC Genomics). In addition, we looked for further genes indicating adaptation to low Mo conditions and found that the PSB MAGs encode high affinity molybdate ABC transporters, which were also transcribed at the time of our study. This is likely an additional adaptation to low Mo conditions. We have included these findings in L. 240-242 and 255-258.

3_ Sequence analysis of *NifD* and *NifK* sequences also suggests that they are phylogenetic distinct from well-studied *NifDK* sequences and perhaps represent an evolutionary representative of ancient nitrogen fixation.

22) Please refer to our answer (21) regarding the phylogeny of *C. okenii* *NifD/NifK*.

4_ Overall Mo-dependent nitrogenase is known to be more predominant and alternative enzymes coded by *vnf* and *anf* systems. The results presented in this study are compatible with the model that Mo-independent systems have emerged later on Earth history.

23) We agree. We have now explicitly stated this in the main text (L. 381-383).

REVIEWERS' COMMENTS

Reviewer #1 (Remarks to the Author):

None

Reviewer #2 (Remarks to the Author):

The authors have addressed all major comments. The revised manuscript reads well and presents very interesting new insights into biogeochemical nitrogen cycling in anoxic environments. It will be a an important contribution to our understanding of past environments. I look forward to seeing it in press.

Best wishes,

Eva Stüeken

Reviewer #3 (Remarks to the Author):

no additional comments

Response to Referees

Purple sulfur bacteria fix N₂ via molybdenum-nitrogenase in a low molybdenum Proterozoic ocean analogue, Philippi et al.

Reviewer #1 (Remarks to the Author):

None

Reviewer #2 (Remarks to the Author):

The authors have addressed all major comments. The revised manuscript reads well and presents very interesting new insights into biogeochemical nitrogen cycling in anoxic environments. It will be an important contribution to our understanding of past environments. I look forward to seeing it in press.

Best wishes,
Eva Stüeken

Reviewer #3 (Remarks to the Author):

no additional comments

We thank all three reviewers for their positive comments and constructive feedback.